



# ClinoformNet-1.0: stratigraphic forward modeling and deep learning for seismic clinoform delineation

Hui Gao[1], Xinming Wu[1], Jinyu Zhang[2], Xiaoming Sun[1], and Zhengfa Bi[1]

[1]School of Earth and Space Sciences, University of Science and Technology of China, Hefei 230026, China
[2]Bureau of Economic Geology, Jackson School of Geosciences, The University of Texas at Austin, Austin, Texas 78758, USA

**Correspondence:** Xinming Wu (xinmwu@ustc.edu.cn)

**Abstract.** Deep learning has been widely used for various kinds of data mining tasks but not much for seismic stratigraphic interpretation due to the lack of labeled training datasets. We present a workflow to automatically generate numerous synthetic training datasets and take the seismic clinoform delineation as an example to demonstrate the effectiveness of using the synthetic datasets for training. In this workflow, we first perform stochastic stratigraphic forward modeling to generate

numerous stratigraphic models of clinoform layers and corresponding porosity properties by randomly but properly choosing initial topographies, sea level curves, and thermal subsidence curves. We then convert the simulated stratigraphic models into impedance models by using the velocity-porosity relationship. We further simulate synthetic seismic data by convolving reflectivity models (converted from impedance models) with Ricker wavelets (with various peak frequencies) and adding real noise extracted from field seismic data. In this way, we automatically generate a total of 3000 diverse synthetic seismic data and the

corresponding stratigraphic labels such as relative geologic time models and facies of clinoforms, which are all made publicly available. We use these synthetic datasets to train a modified encoder-decoder deep neural network for clinoform delineation in seismic data. Within the network, we apply a preconditioning process of structure-oriented smoothing to the feature maps of the decoder neural layers, which is helpful to avoid generating holes or outliers in the final output of clinoform delineation. Multiple 2D and 3D synthetic and field examples demonstrate that the network, trained with only synthetic datasets, works

well to delineate clinoforms in seismic data with high accuracy and efficiency. Our workflow can be easily extended for other seismic stratigraphic interpretation tasks such as sequence boundary identification, synchronous horizon extraction, shoreline trajectory identification and so on.

## 1 Introduction

Seismic stratigraphic interpretation is a crucial step for understanding the evolutionary history information of the sedimentary basin based on seismic data, and is particularly applicable to the sedimentary basin where well data are lacking or unavailable (Nanda, 2021). With the development of seismic data from 2D to 3D and the dramatic increase in the amount of data, the automatic realization of seismic stratigraphic interpretation is the trend. Currently, deep learning has been successfully applied in geophysical interpretation tasks such as fault detection (Huang et al., 2017; Di et al., 2018; Zhao and Mukhopadhyay, 2018;





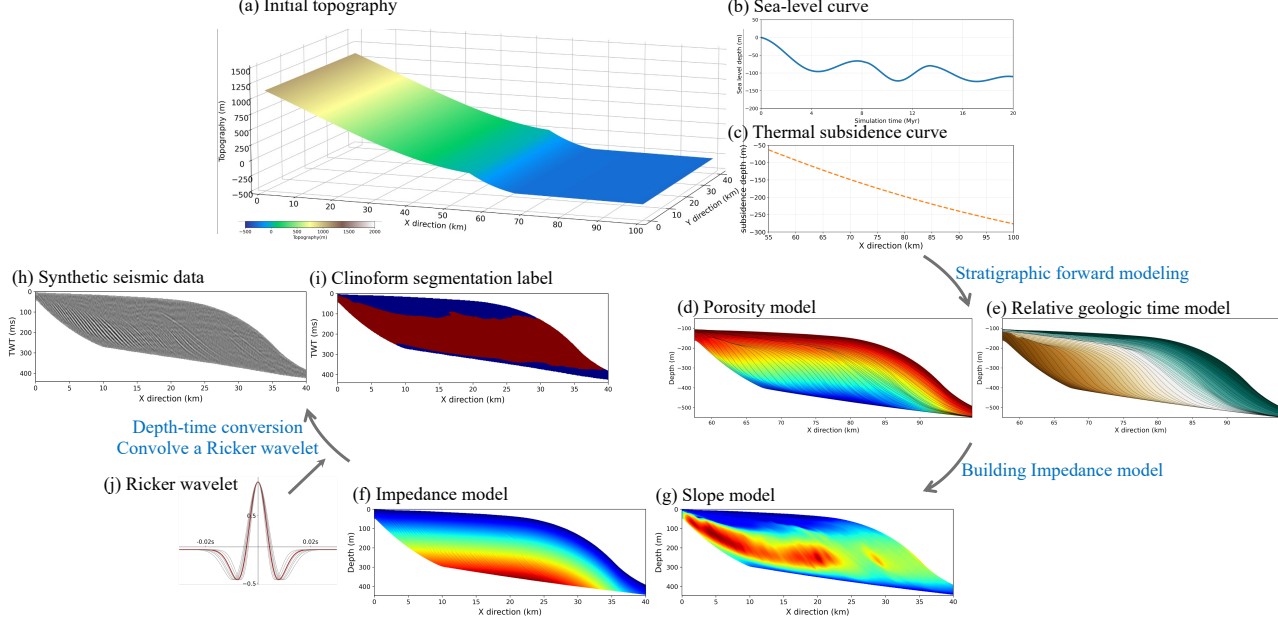

**Figure 1.** Workflow of generating the synthetic clinoform seismic data and corresponding label. We first use a randomly generated (a) initial topography, (b) sea-level curve, and (c) thermal subsidence curve to obtain (d) a porosity model and (e) relative geologic time by stratigraphic forward modeling. Then we use an interpolation method and velocity-porosity relationship to obtain the corresponding (f) P-wave impedance model (g) and slope model. Finally, we obtain the (h) synthetic seismic data and (i) the corresponding segmentation label by depth-time conversion, convolving with (j) a Ricker wavelet with a random peak frequency and adding real noise.

Wu et al., 2019), horizon interpretation (Lowell and Paton, 2018; Wu and Zhang, 2018), salt body delineation (Shi et al., 2019) and so on. However, deep-learning-based automatic seismic stratigraphic interpretation is not discussed much due to the lack of the corresponding training labels. In this work, we take the seismic clinoform delineation task as an example to discuss how to solve the problem of lacking training datasets by stratigraphic forward modeling and further train a deep neural network for efficient and accurate seismic stratigraphic interpretation.

The clinoform and clinothem are widely studied as important sedimentary archives and underpin sequence stratigraphy (Patruno and Helland-Hansen, 2018; Pellegrini et al., 2020). Rich (1951) first introduces the concepts of clinoform and clinothem and proposes the partitioning of deposited surfaces into undaform, clinoform, and fondoform. Steel et al. (2002) redefine clinoform as a deposition surface containing undaform and fondoform, and divide the clinoform into topset, foreset, and bottomset. In this work, we use the term clinoform to describe a surface with a sigmoidal shape, inclined and dipping toward the basin

over a wide range of spatial and temporal scales (Rich, 1951; Bates, 1953; Asquith, 1970; Pirmez et al., 1998; Adams and Schlager, 2000; Steel et al., 2002; Patruno and Helland-Hansen, 2018; Ramon-Duenas et al., 2018), and the term clinothem to describe the clinoform-bounded sedimentary body that records the incremental filling of a sedimentary basin (Patruno and Helland-Hansen, 2018; Ramon-Duenas et al., 2018). In addition, in the field of oil and gas exploration, clinoforms attract the





attention of oil and gas companies in recent years because most of them are low-permeability mudstones or cement, making

them potential reservoir units for hydrocarbon resources.(Cummings and Arnott, 2005; Sydow et al., 2013; Holgate et al., 2013, 2014; Patruno et al., 2018).

The geometry of the clinoform can provide information about paleo-depth, shoreline changes, and the changes in sediment supply and accommodation (Pellegrini et al., 2020). Clinoform can be divided into three components according to geometry. The topset is characterized by low gradient strata (typically < 0.1°) at the top and is dominated by muddy siltstone and fine

to medium-grained sandstone. The foreset is characterized by the steepest slope in the middle and is dominated by muddy siltstone and fine sandstone. The bottomset is characterized by a low gradient toward the basin and is dominated by turbidite sandstone (Patruno and Helland-Hansen, 2018; Ramon-Duenas et al., 2018; Pellegrini et al., 2020).

Currently, the delineation of the three components of the clinoform (topset, foreset, and bottomset) in seismic data is still mainly based on human interpretation by experienced geologists, and therefore remains a labor-intensive task, especially in 3D

seismic volumes. With the recent development of artificial intelligence, deep learning methods have been successfully applied to automate data interpretation tasks in various fields. The convolutional neural network (CNN) method has been proven to be the most powerful method in semantic segmentation (Ronneberger et al., 2015; He et al., 2016; Badrinarayanan et al., 2017; Chen et al., 2018). Recently, the mainstream networks for semantic segmentation are U-net proposed by Ronneberger et al. (2015) and DeepLabV3+ proposed by Chen et al. (2018). However, it is challenging to apply the CNN method in solving

geoscience problems including the seismic stratigraphic interpretation because of the lack of a large amount of training data sets and the related labels (Bergen et al., 2019; Di et al., 2020).

To solve this problem, we consider performing numerical stratigraphic forward modeling (SFM) to generate numerous synthetic datasets with labels. SFM has been widely used in the study of sequence stratigraphy over the last decades (Martin et al., 2009; Burgess et al., 2012; Sylvester et al., 2015; Harris et al., 2016). Many mathematical and physical models have

been proposed to simulate the evolution of sedimentary basins by using various numerical algorithms to obtain synthetic stratigraphic states in full consideration of thermal subsidence, uplift, changes in sediment supply, various sediment transport and deposition processed (Warrlich et al., 2008; Shafie and Madon, 2008; Burgess et al., 2012; Hawie et al., 2015; Salles et al., 2018). While most of the previous work performs SFM to fit the stratigraphic model of a specific basin by carefully choosing a set of modeling parameters, we perform stochastic SFM to generate numerous models by randomly choosing various sets of

parameters.

In this paper, we develop a workflow (Fig. 1) to automatically generate synthetic seismic data and corresponding labels and use the synthetic datasets to train a CNN for seismic clinoform delineation. We first use a numerical SFM method, modified from pyBadlands (Salles et al., 2018) to generate a synthetic porosity model (Fig. 1d) and relative geologic time model (Fig. 1e) of clinoform layers with inputs of a randomly but properly chosen initial topography surface (Fig. 1a), sea-level changes

(Fig.1b), and thermal subsidence curve (Fig. 1c). Then we use an interpolation method and a velocity-porosity relationship proposed by Krief et al. (1990) to build an impedance model (Fig. 1f) and at the same time calculate a slope volume (Fig. 1g) from the corresponding depth model of clinoform layers. Finally, we perform depth-to-time conversion to the impedance model, convolve it with a Ricker wavelet with a random peak frequency (Fig.1j) and add real noise to generate a synthetic seis-





mic image (Fig. 1h). At the same time as generating the synthetic seismic data, we also automatically obtain the corresponding
stratigraphic labels of relative geologic time and clinoform delineation (Fig. 1i). After generating various clinoform seismic
data and corresponding labels, we use them to train a CNN modified from DeepLabv3+ for automatic clinoform delineation in
seismic data. We demonstrate the performance of the trained network on both synthetic and field seismic data.

## 2 Training data generation

In this section, we implement a workflow of geological and geophysical forward modeling process (Fig. 2) to automatically
generate diverse synthetic clinoform seismic data and corresponding labels to train a CNN for seismic stratigraphic interpreta-
tion. The black boxes in Fig. 2 contain the SFM process based on the pyBadlands (Salles et al., 2018), and the blue boxes in Fig.
2 contain the geophysical forward modeling processes of generating synthetic seismic data and corresponding segmentation
labels. In this workflow, we first randomly generate diverse SFM inputs, such as initial topography, sea level curve, rainfall pat-
terns, uplift, subsidence, etc. We then simulate numerous stratigraphic models of clinoform layers using stratigraphic forward
modeling with these diverse model inputs. After obtaining these stratigraphic models of clinoform layers, we perform an inter-
polation method and a velocity-porosity relationship to build the velocity and impedance models. We further use the velocity
model generated in the previous step to convert the stratigraphic models from the depth domain to the time domain. Finally, we
convolve the reflectivity model (converted from the impedance model) with a Ricker wavelet to generate the synthetic seismic
data and add real noise to improve the realism of the synthetic seismic data.

### 2.1 Stratigraphic forward modeling

Many SFM methods have been proposed to simulate the evolution of sedimentary basins by using various numerical algorithms
to obtain synthetic clinoform layers in full consideration of thermal subsidence, uplift, changes in sediment supply, various
sediment transport and deposition processed (Warrlich et al., 2008; Shafie and Madon, 2008; Burgess et al., 2012; Hawie et al.,
2015; Salles et al., 2018). In this study, we use the SFM method implemented in the pyBadlands (Salles et al., 2018) to simulate
numerous stratigraphic models of clinoform layers. Pybadlands is a long-term surface evolution model that simulates sediment
transport and deposition from source to sink (Salles and Hardiman, 2016; Salles et al., 2018; Ding et al., 2019).

At each time step of the simulation, the sedimentary structure of the clinoform layers is mainly controlled by the rate of
accommodation change $(\delta A)$ and the rate of sediment-supply change $(\delta S)$. When $\delta S \geq \delta A$, clinothem typically accumulates
and moves toward the sedimentary basin, and when $\delta S \leq \delta A$, clinothem typically moves and degrades toward the continental
shelf (Patruno and Helland-Hansen, 2018; Pellegrini et al., 2020). The rate of accommodation change $(\delta A)$ reflects the size of
the available space accommodated sediments and is mainly related to thermal subsidence and sea level fluctuations, etc. The
rate of sediment-supply change $(\delta S)$ is mainly related to rainfall patterns, topography, rock erodibility, etc (Schlager, 1993;
Muto and Steel, 1997; Hawie et al., 2015; Neal et al., 2016).



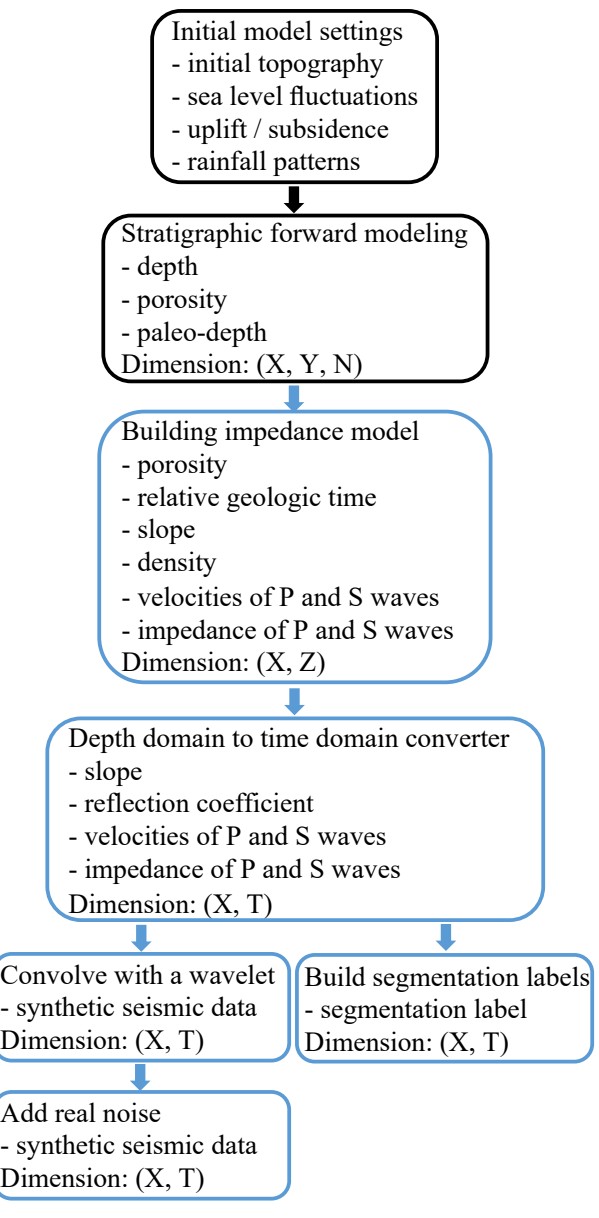

**Figure 2.** The workflow is used to generate diverse synthetic clinoform seismic data and corresponding segmentation labels. The black boxes contain the steps of the stratigraphic simulation process in pyBadlands (Salles et al., 2018). X and Y represent the length of inline and crossline directions, respectively. N represents the number of clinoform layers generated in the simulation. The blue boxes contain the process of generating synthetic clinoform seismic data and corresponding segmentation labels. Z represents the vertical magnitude of the interpolated formation attributes in the depth domain, and T represents the vertical magnitude of the interpolated formation attributes in the time domain.





**Figure 3.** Some examples of (a) initial topography, (b) eustatic sea level curves, and (c) thermal subsidence curves. Initial topography consists of a 55 km ($x$ direction) by 40 km ($y$ direction) mountain range (erosion region) and a 45 km ($x$ direction) by 40 km ($y$ direction) sedimentary basins (sedimentary region). The average slope of the mountain range varies from $1.0°$ to $1.3°$. Sedimentary basins contain a slope with an average slope of $1.2°$ to $1.5°$ and a basin with an average slope of about $0.03°$. The sea level curve consists of multiple sets of sinusoids whose periods are randomly set and depths vary randomly between $50m$ and $-200m$. The thermal subsidence curve of the sedimentary basin is determined by a distance-dependent stretching factor (Salles et al., 2018; Ding et al., 2019), which is randomly set to vary from 1.20 to 1.58.



**Table 1.** Summary of initial parameter ranges for stratigraphic forward modeling

| Initial topography | Mountain range | Height | $1000m$-$1200m$ |
| | | average slope | $1.0°$ to $1.3°$ |
| | Sedimentary basin | Depth | $200m$-$300m$ |
| | | average slope | $1.2°$ to $1.5°$ and $0.03°$ |
| Sea level curve | Period | | $4Myr$ - $20Myr$ |
| | Amplitude | | $-200m$ - $50m$ |
| Subsidence | | | $2.48m \cdot Myr^{-1}$ - $15.0m \cdot Myr^{-1}$ |
| Rainfall patterns | | | $0.5m \cdot yr^{-1}$ - $4m \cdot yr^{-1}$ |

The whole workflow of simulating numerous stratigraphic models of clinoform layers is summarized in the black boxes in Fig. 2. In this simulation process, we generate 200 clinoform layers for each model and set the grid size and time step as 100m×100m and 0.1 Myr, respectively. Besides, in order to obtain diverse geometric clinoform layers, the initial topography, sea level curve, thermal subsidence, and rainfall patterns are randomly chosen from some predefined ranges in Table. 1.

As shown in Fig. 3a, the initial topography for simulation has a spatial scale of 100 km ($x$ direction) by 40 km ($y$ direction) and with a grid size of 100 m by 100 m. The mountain range (erosion region) is 55 km ($x$ direction) by 40 km ($y$ direction) while the sedimentary basin (sedimentary region) is 45 km ($x$ direction) by 40 km ($y$ direction). We randomly set the highest altitude of the mountain range within 1000 to 1200 m and set the average slope in the range between 1.0 and 1.3 degrees. The sedimentary basin is divided into two parts: an inclined basin margin region and a relatively flat basin center region. 115 The average slope of the inclined basin margin area and the flat basin center area is set to 1.2-1.5 degrees and 0.03 degrees, respectively. In this paper, we perform 3D stratigraphic forward modeling to build 3D stratigraphic models from which we extract 2D profiles in the middle of $y$ dimension to build a large training dataset. Therefore, we set a relatively narrow width in the $y$ direction to save computational time and memory in the stochastic simulation of many models.

However, the width of the initial topography should not be too narrow compared to its significantly long extension (100km) 120 in the $x$ direction. Otherwise, a large amount of sediment, produced in the mountain range (erosion region), will spill out of the simulation zone during long-distance transportation to the sedimentary basin, which may result in an unstable simulation and yield unreasonable depositional hiatus in the sedimentary basin. With these considerations, we set the width of the initial





topography to 40km ($y$ direction) to speed up the simulation process and save computational costs while ensuring the stability of the deposition simulation. By randomly choosing the parameters of the mountain range and sedimentary basin as discussed

above, we generate 200 unique initial topographies and we display one of them in Fig. 3a.

The change of sea level is a major control on the accommodation of the sedimentary basin. We randomly generate 1000 different eustatic sea level curves with a period of 20 Myr starting from 0 m, each curve consists of multiple sets of sinusoids. The periods and amplitudes of their sinusoids are randomly set from 4Myr to 20Myr, -200m to 50m, respectively. These randomly generated sea-level curves (Fig. 3b) help simulate diverse stratigraphic models with various sedimentary geometries.

Thermal subsidence can also affect the accommodation of the sedimentary basin in simulation. We use a simple subsidence model and set its corresponding parameters after McKenzie (1978) to simulate the thermal subsidence of sedimentary basins. We also assume that the stretching factor in the model is distance-dependent (Ding et al., 2019), and randomly set the slope of the linear relationship between the stretching factor and distance. Finally, the thermal deposition rate in the sedimentary basin is set to be about $2.48m \cdot Myr^{-1}$ to $15.0m \cdot Myr^{-1}$ (Fig. 3c).

Rainfall patterns can affect sediment supply changes ($\delta S$) because precipitation can erode the mountain range and transport the sediments to the sedimentary basin. We first randomly divide the entire simulated cycle into several parts, and then randomly set spatially uniform precipitation rate (about $0.5m \cdot yr^{-1}$ to $4m \cdot yr^{-1}$) for each part. Besides, in order to guarantee a stable sediment supply within the mountain range during the long-period erosion, we keep uplifting the mountain range while the simulation proceeds. We also predefined an initial surface porosity and then calculated the porosity distribution for

each stratigraphic layer during the simulation according to the porosity-depth relationship suggested by Athy (1930). The porosity-depth relationship takes into account the mechanical compaction of the sediment, i.e., porosity gradually decreases with increasing overburden.

By using various combinations of the 200 initial topographies, 1000 eustatic sea level curves, 30 thermal subsidence curves, different rainfall patterns, etc, we simulate 1000 diverse stratigraphic models, each with multiple clinoform attribute layers

such as depth, relative geologic time, porosity, paleo depth (related to the depositional environment) and so on. Fig. 4 shows 2D slices that are extracted from three of the simulated 3D models. The SFM discussed above actually simulates only the stratigraphic layers denoted by the black curves in Fig. 4. Based on the simulated models of stratigraphic layers or curves, we need to further define the attributes of relative geologic time, paleo-depth, and porosity on the curves and then interpolate the corresponding full models as shown in color in each column of Fig. 4. In the following section, we discuss in detail how to

build such property models and perform geophysical forward modeling to construct training datasets of seismic images and the corresponding stratigraphic labels. With the geological forward modeling process discussed above, we obtain 1000 numerous stratigraphic models of clinoform layers.

## 2.2 Building impedance model

Before generating numerous synthetic seismic data and corresponding segmentation labels, we need to compute impedance and slope models from the porosity and depth models of clinoform layers, respectively. In this study, we use the velocity-



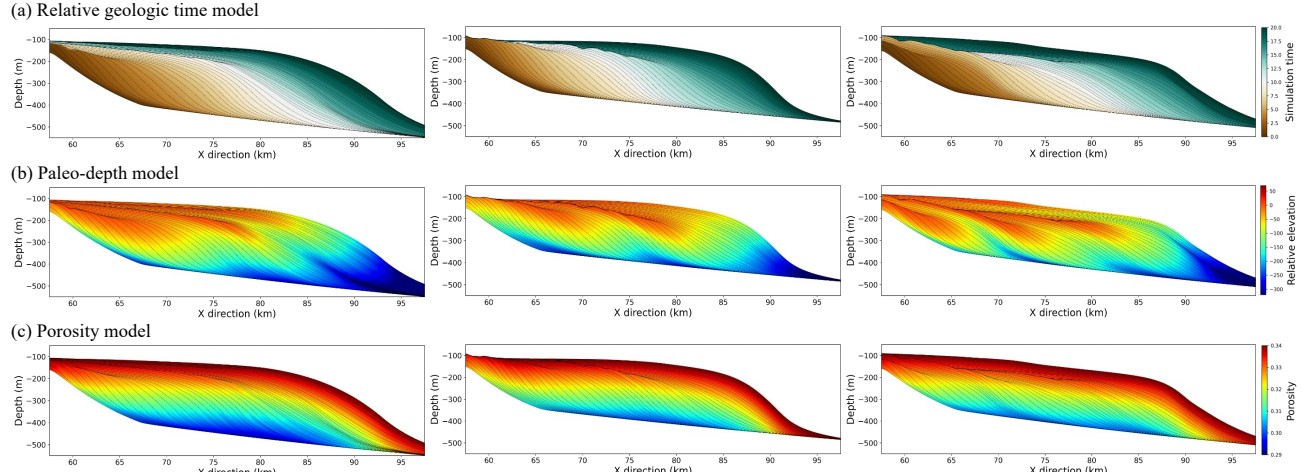

**Figure 4.** Three 2D examples of stratigraphic models with clinoform attribute layers from stratigraphic forward modeling. (a) The relative geologic time model. (b) Paleo-depth model. (c) Porosity model. The dimension of the outputs is (X, Y, N), where X and Y are the length and width of the initial topography with a resolution of 100m, and N is the number of clinoform layers generated during the simulation. The black curves in the figure represent the positions of each clinoform attribute layer displayed at 4 Myr intervals. The colors filled in the figures represent the simulation time, paleo depth, and porosity, respectively.

porosity relationship model proposed by Krief et al. (1990) to build impedance models, which assumes that the velocities of P and S waves in porous fluid-saturated rocks obey Gassmann equations with the Biot compliance coefficient (Krief et al., 1990; Gassmann, 1951; Biot, 1941; Goldberg and Gurevich, 2008). To obtain an impedance model, we first calculate the formation

density $\rho_{fm}$, shear modulus $\mu_{fm}$ and bulk modulus $K_{fm}$, by using the following equations suggested by Krief et al. (1990), Lee (2005) and Gassmann (1951):

$$\rho_{fm} = (1-\phi)\rho_{ma} + \phi\rho_{fl}, \tag{1}$$

$$\mu_{fm} = \mu_{ma}(1-\beta), \tag{2}$$

$$K_{fm} = K_{ma}(1-\beta) + \beta^2 M. \tag{3}$$

In the formation density equation (Eq. 1), $\phi$ is porosity values, $\rho_{ma} = 2.65 kg \cdot m^{-3}$ is the density of matrix material and $\rho_{fl} = 1.04 kg \cdot m^{-3}$ is the density of fluid. In the formation shear modulus equation (Eq. 2), $\mu_{ma} = 13.48 Gpa$ is matrix material shear modulus and $\beta$ is Biot coefficient, and it is assumed that the presence of fluid has no or little influence on the shear modulus of the formation (Krief et al., 1990; Lee, 2005). The Biot coefficient $\beta$ is related to the porosity and can be calculated by the following empirical relationship equations suggested by Krief et al. (1990):

$$(1-\beta) = (1-\phi)^{m(\phi)}, \tag{4}$$

$$m(\phi) = \frac{3}{(1-\phi)}. \tag{5}$$



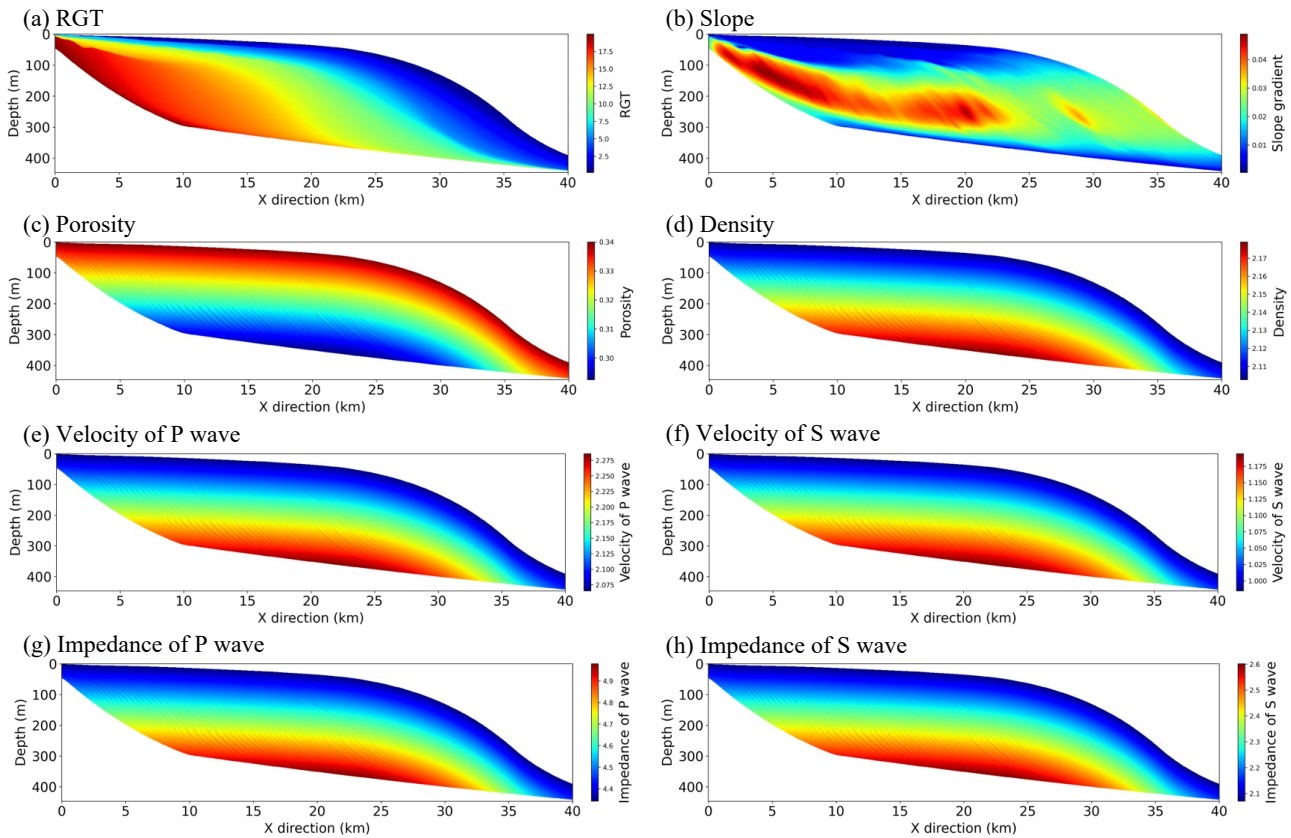

**Figure 5.** Examples of 2D clinoform attribute models obtained by an interpolation method and velocity-porosity relationship. (a) Relative geologic time (RGT). (b) Slope. (c) Porosity. (d) Density. (e) P-wave velocity. (f) S-wave velocity. (g) P-wave impedance. (h) S-wave impedance. The dimension of the outputs is (X, Z), where X is the length of the 2D clinoform attribute model with a resolution of 25m, and Z is the depth of the 2D clinoform attribute model with a resolution of 1m.

In the empirical relationship equations (Eq. 4 and Eq. 5), $m(\phi)$ is a function of the porosity. In the formation bulk modulus equation (Eq. 3), $K_{ma} = 15.45 GPa$ is matrix material bulk modulus, $K_{fl} = 2.25 GPa$ is fluid bulk modulus and $M$ is a modulus which is dependent on the Biot coefficient and can be calculated by the following equation suggested by Gassmann

175 (1951):

$$\frac{1}{M} = \frac{(\beta - \phi)}{K_{ma}} + \frac{\phi}{K_{fl}}. \tag{6}$$

In these equations (Eq. 1 to Eq. 6), some predefined parameters are taken from Carcione et al. (2002). After calculating the $K_{fm}, \rho_{fm}, \mu_{fm}$, we can calculate the velocity of P and S waves, the impedance models of P and S waves by the following





equations:

$$V_P = \sqrt{\dfrac{K_{fm} + \dfrac{4}{3}\mu_{fm}}{\rho_{fm}}}, \tag{7}$$

$$V_S = \sqrt{\dfrac{\mu_{fm}}{\rho_{fm}}}, \tag{8}$$

$$Z_P = \rho_{fm} \cdot V_P, \tag{9}$$

$$Z_S = \rho_{fm} \cdot V_S, \tag{10}$$

where $V_P$ is the P-wave velocity, $V_S$ is the S-wave velocity, $Z_P$ is the P-wave impedance, $Z_S$ is the S-wave impedance. At the same time, we further use the depth model of clinoform layers to calculate the slope model used to generate the segmentation label.

After calculating these stratigraphic and seismic attribute models of clinoform layers, we also need to perform the interpolation method to build corresponding 2D attribute models. Because the dimensions of these stratigraphic and seismic attribute models of clinoform layers are not length ($x$-direction) and depth ($z$-direction) but length ($x$-direction) and the number of stratigraphic clinoform lines ($n$). Besides, the resolution of the inline ($x$-direction) direction is generally 25m in field seismic data, but the resolution of each stratigraphic clinoform line in the inline direction ($x$-direction) is 100 m. We first interpolate these stratigraphic and seismic attribute models of clinoform layers in inline directions ($x$-direction) to match the resolution of field seismic data. Then we convert then (X, N) into corresponding 2D attribute models of dimension (X, Z) by interpolation in the depth direction ($z$-direction) (see Fig. 5).

### 2.3 Building synthetic seismic data and corresponding segmentation labels

In practice, the field seismic data are usually in the time domain. In this case, to be consistent with the field seismic data, we need to convert these 2D clinoform attribute models from the depth domain to the time domain by the depth-to-time relationship as follows:

$$t[i] = t[i-1] + 2 \cdot \dfrac{dz}{v_p[i]}, \tag{11}$$

where $t[i]$ and $t[i-1]$ represent two-way travel time, $dz$ is the vertical sampling rate in the depth domain (i.e. the distance between the $(i-1)$-th and $i$-th samples), $v_p[i]$ is the velocity of P wave at the $i$-th sample. By performing the iterative operation on each column of the two-dimensional velocity model and resampling according to a predefined time sampling rate, we can obtain the mapping relationship from the depth domain to the time domain. Then we generate the stratigraphic and seismic attribute models of dimensions (X, T) by applying this calculated mapping relationship.

To generate the synthetic seismic data, we convolve the reflectivity model (converted from the impedance model) with a Ricker wavelet with a random peak frequency shown in Fig. 1j. The peak frequency of the Ricker wavelet is randomly chosen



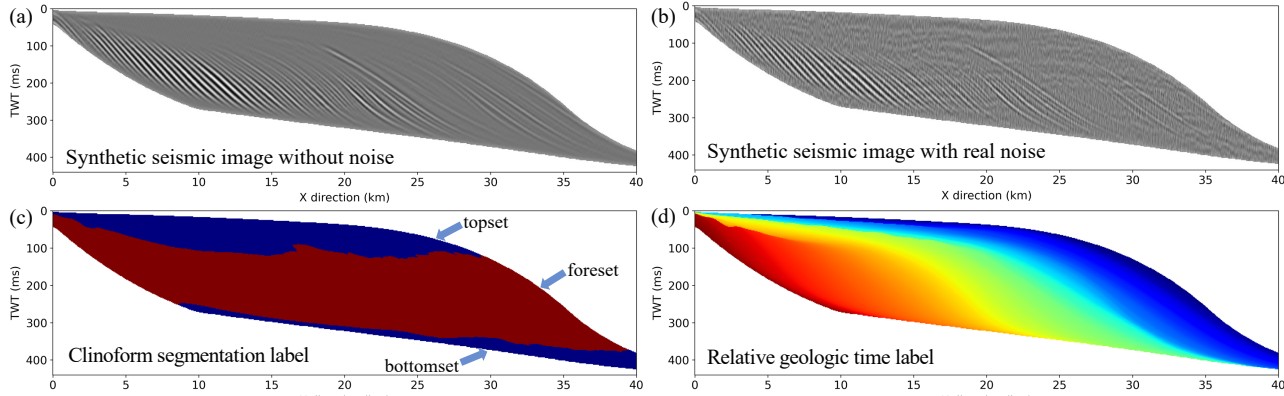

**Figure 6.** Example of (a) synthetic seismic image without noise, (b) synthetic seismic image with noise extracted from field seismic data, (c) corresponding clinoform segmentation label, and (d) relative geologic time label. The foreset part of the clinoform is filled in red, whereas the topset and bottomset parts of the clinoform are filled in blue. The dimension of the profiles is (X, T), where X is the length of the 2D profiles with a resolution of 25m, and T is the two-way travel time of the 2D profiles with a resolution of 2ms.

from a predefined range ($40Hz - 60Hz$) with reference to the peak frequency extracted in the field seismic data (Tetyukhina et al., 2010). Fig. 6a shows a synthetic clinoform seismic profile simulated in this way.

To further improve the realism of the synthetic seismic data, we first extract the real noise from the filtered seismic data of the F3 block in the North Sea, Australia Poseidon, and Alaska North Slope data. Then we perform data augmentation on different noise data by flipping and synthesizing all the enhanced real noise data into a 3D noise volume. Finally, we add 2D real noise (with randomly chosen a slice from 3D noise volume) to the 2D seismic data according to a randomly selected signal-to-noise ratio and we display the synthetic seismic data with real noise in Fig. 6b.

Finally, we generate the corresponding segmentation label using the slope model in the time domain. We set the region with a slope greater than 0.0175 (slope angle $> 1°$) as the foreset areas of the clinoform, and other regions as the topset and bottomset areas of the clinoform. The threshold value of the slope refers to Pellegrini et al. (2020). The corresponding clinoform label and relative geologic time label are shown in Fig. 6c and Fig. 6d, respectively.

In addition, we can automatically generate numerous and diverse synthetic seismic data and the corresponding clinoform
attribute models (e.g. slope model, sea level curve, relative geologic time model, paleo-depth model, etc.) based on these geological and geophysical forward modeling processes. This indicates that our workflow can be easily extended for other seismic stratigraphic interpretation tasks such as clinoform delineation, sequence boundary identification, synchronous horizon extraction, shoreline trajectory identification, sedimentary facies analysis and so on. In this work, we take the seismic clinoform delineation task as an example to demonstrate the effectiveness of using the synthetic dataset for training.






## 3 Deep learning for clinoform delineation

We consider clinoform delineation as an image segmentation problem with the goal to label ones on the foreset part of the clinoform, whereas zeros on the topset and bottomset parts of the clinoform in 2D seismic data. In this study, we use a modified encoder-decoder deep neural network to implement the clinoform delineation. We first train the network with synthetic seismic

data and corresponding segmentation labels and then use the trained network to delineate clinoform in both synthetic and field seismic data.

### 3.1 Network architecture

As shown in Fig. 7, the architecture of our encoder-decoder deep neural network used in this work is modified from DeepLabv3+,

which was proposed by Chen et al. (2018). It has achieved state-of-the-art performance in the natural image semantic segmentation field (PASCAL VOC 2012 and Cityscapes datasets). The backbone of the DeepLabv3+ is an encoder-decoder architecture, which downscales the inputs images and extracts low-level semantic features and high-level semantic features in the encoder module and gradually recovers sharp boundaries in the decoder module (Ronneberger et al., 2015; Badrinarayanan et al., 2017; Chen et al., 2018). However, compared to the original DeepLabv3+, we remove the skip connections between the encoder and

decoder layers, because we found that the low-level features from the encoder layers lead to some artifacts in the predicted results.

In the modified network, the encoder module (see left gray dashed box in Fig. 7) consists of a deep convolutional neural network (DCNN) and an atrous spatial pyramid pooling module (ASPP). The DCNN module consists of a ResNet-101 (He et al., 2016) and is used to progressively reduce the resolution of feature maps and capture high-level semantic features. ResNet-

101 consists of a $7 \times 7$ convolutional layer with a stride of 2, a $3 \times 3$ max-pooling with a stride of 2, and four ResBlocks. These four ResBlocks are composed of bottleneck blocks with the number of 3, 4, 23, 4, and strides of 1, 2, 1, and 1, respectively. The bottleneck block consists of a $1 \times 1$ convolutional layer, a $3 \times 3$ convolutional layer, a $1 \times 1$ convolutional layer, and a skip connection (see blue box in Fig. 7). The ASPP module can capture the contextual information at multiple scales, which are composed of a $1 \times 1$ convolutional layer, three $3 \times 3$ atrous convolutional layers with sampling strides of 6, 12, and 18,

and an average pooling module (see the orange box in Fig. 7).

The decoder module in the modified networks (see right gray dashed box in Fig. 7 ) first uses a bilinear upsampling layer to upsample the high-level semantic features by a factor of 4, and then the upsampled features are fed to two $3 \times 3$ convolutional layers and a $1 \times 1$ convolutional layer which obtains the probabilities of each class and reduces the channels of features to the number of classes. Finally, the same bilinear upsampling layer is applied to obtain the final feature maps of the same size as the

input image. These final feature maps are fed to a softmax layer and an argmax layer to obtain the final clinoform segmentation result.



**Figure 7.** The architecture of the network for clinoform delineation. The left gray dashed box is the encoder module which consists of a ResNet-101 and an ASPP module. The blue box in the middle is the module of ResBlocks in the encoder, which contains a $1 \times 1$ convolutional layer, a $3 \times 3$ convolutional layer, a $1 \times 1$ convolutional layer, and a skip connection. The orange box in the middle is the module of ASPP in the encoder, which contains a $1 \times 1$ convolutional layer, three $3 \times 3$ atrous convolutional layers with sampling strides 6, 12, and 18, and an average pooling module. The right gray dashed box is the encoder module which consists of an upsampling layer, two $3 \times 3$ convolutional layers, a $1 \times 1$ convolutional layer, and an upsampling layer.

## 3.2 Training and validation

After automatically generating 1000 pairs of synthetic seismic data (Fig. 6b) and corresponding clinoform labels (Fig. 6c), we
train and validate our CNN model by using 800 and 200 pairs of them, respectively. Considering the amplitude values of field seismic data can vary widely, we perform a normalization process to each seismic data before feeding it into the CNN model by subtracting by its mean value and then dividing by its standard deviation.





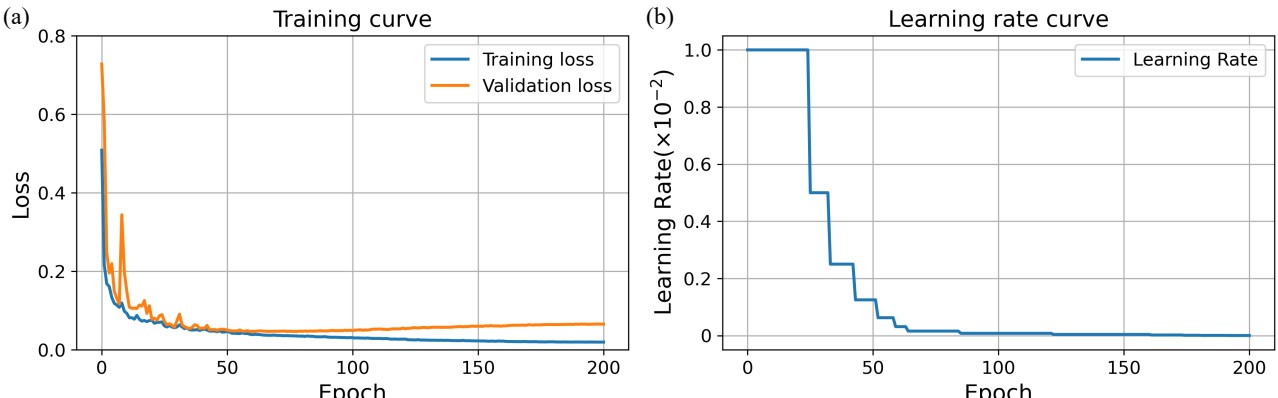

**Figure 8.** (a) The loss curves of training (blue) and validation (orange), (b) and the learning rate is adaptively adjusted in the training process. The curves for both training (blue curve) and validation (orange curve) losses converge to 0.02 and 0.06, whereas the learning rate decrease to 0.00001 after 200 epochs.

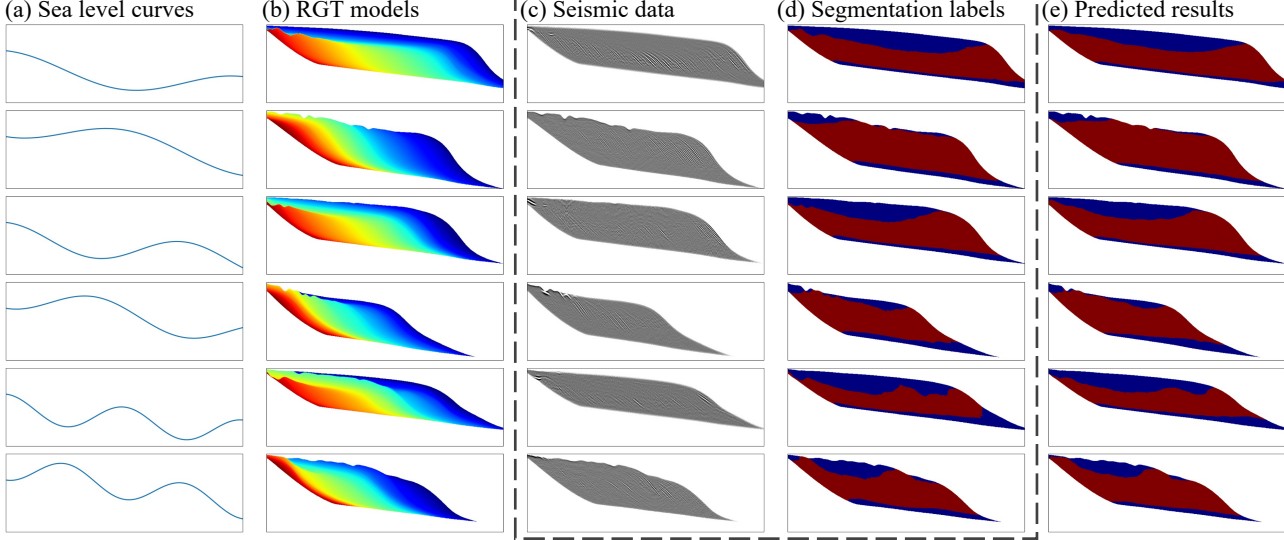

**Figure 9.** Examples of (a) classic sea level curves with 1, 1.5, and 2 cycles, (b) relative geologic time model (RGT), (c) synthetic seismic image with real noise, (d) corresponding segmentation labels, and (e) predicted results using the trained network. The foreset part of the clinoform is filled in red, whereas the topset and bottomset parts of the clinoform are filled in blue. From the predicted results, we can find that (e) the predicted results are consistent with (d) the labels, which means the network has successfully learned to extract the structural features for clinoform segmentation in the seismic images.

Considering clinoform delineation in the 2D seismic image is a binary segmentation problem, we use the following binary cross entropy loss function $\mathcal{L}$ to train our network:

$$\mathcal{L} = -\sum_{i=1}^{N} y_i log(x_i) - \sum_{i=1}^{N} (1 - y_i) log(1 - x_i), \tag{12}$$






where $N$ denotes the number of pixels in the input 2D seismic data. $x_i$ and $y_i$ represent the prediction and label at the $i-$th pixel , respectively.

In the training of the network, the size of each synthetic seismic data and corresponding segmentation label is $1600 \times 256$ pixels. Considering the computation time and memory, we feed the normalized synthetic seismic images and corresponding

segmentation labels to the CNN in batches and set the batch size to 16. We use the Adam optimizer (Kingma and Ba, 2014) to optimize the network parameters. In the training process, we start the learning rate at 0.01 and adaptively reduce it based on the validation loss. Specifically, we automatically reduce the learning rate by half when the validation metric stagnates within 2 epochs. In total, we use 800 and 200 pairs of synthetic training datasets to train and validate our CNN model. As shown in Fig. 8, the curves for both training (blue curve) and validation (orange curve) losses converge to 0.02 and 0.08, while the learning

rate decreases to 0.00001 after 200 epochs.

To demonstrate the performance of the trained network, we first apply it to six synthetic seismic images with classic clinoforms (Fig. 9c) that are generated by six classic sea-level curves with 1, 1.5, and 2 sinusoidal periods as shown in Fig. 9a. Note that real noise extracted from field seismic data has been added to the synthetic seismic data (Fig. 9c). By feeding the synthetic seismic data to our trained network, we obtain the clinoform segmentation results shown in Fig. 9e. We observe

that the segmentation results are consistent with the ground truth labels shown in Fig. 9d. It is not surprising that the trained network works well in these synthetic seismic data simulated by using the same workflow of generating the training dataset. We therefore further validate our network in multiple field seismic data in the following sections.

### 3.3   Eliminating outliers by smoothing feature maps

We further verify the performance of our trained network on a field seismic data with more complex features as shown in Fig. 10a. Fig. 10b shows the predicted result, where red and blue represent the foreset part and topset or bottomset parts of the clinoform, respectively. We observe that most of the foreset areas are correctly predicted, but the areas indicated by the cyan arrows are incorrectly predicted as topset or bottomset of the clinoform, and the area indicated by the green arrow is incorrectly predicted as the foreset part of the clinoform.

To better understand the feature extraction and prediction process of the network in segmenting a clinoform, we display the feature maps of each layer in the network in Fig. 11. The encoder module in the networks (see the left part in Fig. 11) focuses on extracting features of the foreset part of the clinoform, topset or bottomset parts of the clinoform, and boundaries. The decoder module in the network (see the right part in Fig. 11) focuses on gradually recovering the specific areas of the foreset part of the clinoform and topset or bottomset parts of the clinoform according to the features extracted by the encoder

module. We expect that the foreset part of the clinoform should be a complete and continuous block, however, the predicted result shown in Fig. 10b obviously does not conform to such a priori geological understanding.

In order to make the segmentation results of the trained network on the field seismic data more clean and complete, we introduce a layer of structure-oriented smoothing into the network. This smoothing layer enhances the structural features along the seismic structural orientations, thus filling holes and eliminating outliers in the final segmentation results. We first estimate





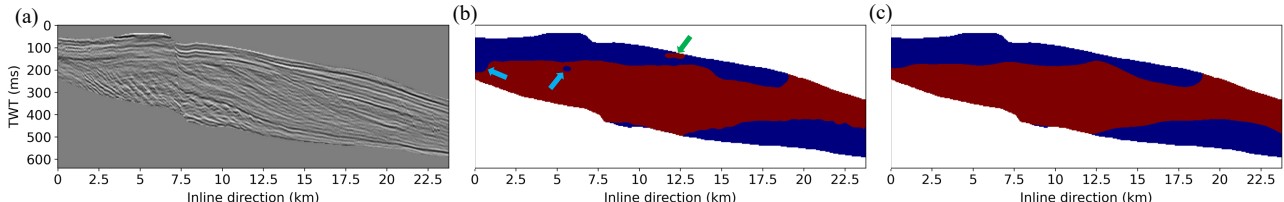

**Figure 10.** We input (a) a field seismic profile to input into the trained network and get (b) the corresponding predicted result, where red represents the foreset part of the clinoform and blue represents the topset and bottomset parts of the clinoform. The clinoform segmentation result is mostly correct but we also observe some holes and outliers as indicated by cyan and green arrows in (b), respectively. To further fill the holes and eliminate the outliers, we introduce structural smoothing layers into the network to enhance the feature maps of the network and therefore obtain a cleaner and more complete result in (c).

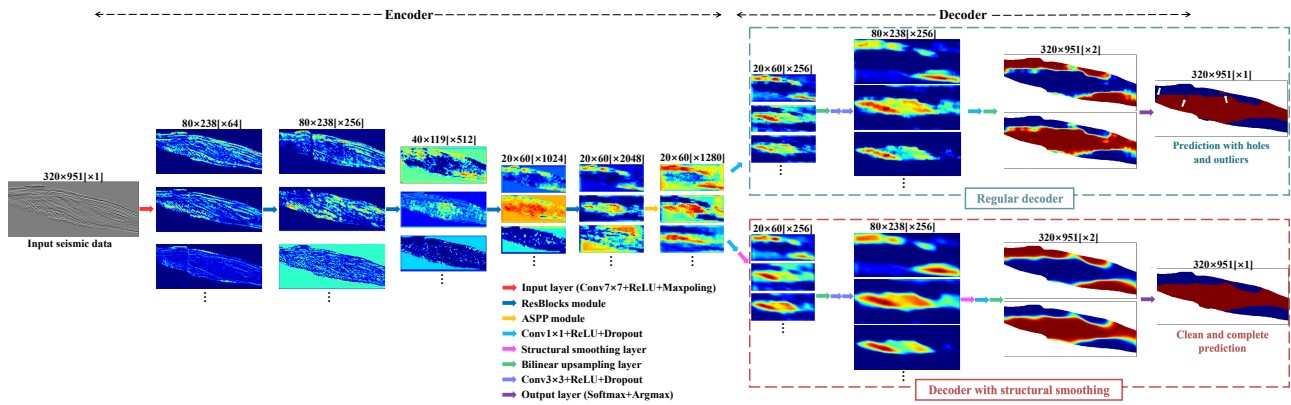

**Figure 11.** The feature maps of the encoder-decoder networks. The red arrow denotes an input layer containing a $7 \times 7$ convolutional layer with ReLU and a max-pooling layer. The navy blue arrow denotes four different ResBlocks modules consisting of bottleneck blocks with numbers of 3, 4, 23, 4, and strides of 1, 2, 1, and 1, respectively. The yellow arrow denotes the ASPP modules. The light blue arrow denotes a $1 \times 1$ convolution layer with ReLU and dropout. The pink arrow denotes structural smoothing layers at 1/4 or 1/16 scale. The green arrow denotes a bilinear upsampling layer. The light purple arrow denotes a $3 \times 3$ convolution layer with ReLU and dropout. The navy purple denotes an output layer consisting of a softmax layer and an argmax layer. The upper right green dashed box and lower right red dashed box refer to the feature maps and predicted results before and after the introduction of the structural smoothing layer, respectively.

the structural orientation at each pixel of the input seismic image by using the method of structure tensor (Van Vliet and Verbeek, 1995; Fehmers and Höcker, 2003; Hale, 2009). Then we construct the corresponding smooth convolution kernels along the seismic structures based on the estimated orientations. Finally, we apply these structure-oriented smoothing kernels to the feature maps at the scales of 1/4 and 1/16 in the network, as denoted by pink arrows in Fig. 11.

After introducing the structural smoothing layer into the network, the predicted results shown in Fig. 10c are more complete and continuous. To visually demonstrate the function of the introduced structural smoothing layers, we display the feature maps





of the decoder module after introducing the structure-oriented smoothing layers in the dashed red box in Fig. 11. Compared to the original decoder module in the dashed green box, the feature maps with the smoothing layers become cleaner and more continuous for the features of topset, foreset, and bottomset of the clinoform, and the predicted result based on these feature maps is therefore cleaner and more complete.


## 4    Field data applications

In addition to the synthetic seismic data, we also use three different field seismic data to verify the performance of the network trained with only synthetic seismic data. In our simulated synthetic training datasets, the seismic images (like the one in Fig. 6b) contain only the simulated processes of the clinoform deposition with dramatic changes in sea level but do not contain the

non-clinoform part with horizontal deposition. However, the field seismic images typically contain a large number of horizontal deposits, and the clinoform deposits are only a small part of them. In order to be consistent with the training data sets, we first use an automatic horizon picking method proposed by Wu and Fomel (2018) to extract the top and bottom layers of the part of clinoform deposition in the field seismic data. Then we cut out the sub-volume with the clinoform deposits from the whole seismic volume by using the computed two boundary layers. To avoid the uncertainty of the seismic amplitude variation range of the field survey seismic data, each of them is first subtracted by its mean and then divided by its standard deviation to obtain

a normalized seismic data.

### 4.1    Case study one: Netherlands offshore F3 Block

The first field example is the Netherlands offshore F3 Block (Dutch sector) seismic data acquired in the northeastern part of

the North Sea. In this survey, a large fluvio-delta-ic system dominated the basin resulting in a massive clinoform deposition during the Cenozoic era (Tetyukhina et al., 2010). The survey area of this seismic volume is 24km in the inline direction, 16km in the crossline direction, and 1848ms in the two-way travel time direction. In this work, we select a 3D subset (951[inline] × 460[crossline] × 320[time] samples) of F3 Block seismic data as the study seismic volume (Fig. 12a).

We first use the method proposed by Wu and Fomel (2018) to extract the top and bottom surfaces of the part of clinoform

deposition in this seismic volume and then cut the sub-volume (Fig. 12a) between the two surfaces. We then apply the trained network to delineate the clinoform in the 3D sub-volume slice by slice in the crossline direction. Each 2D slice (951[inline] × 320[time] samples) is normalized by its mean and standard deviation before fed into the network. In these seismic images, some complex geological structures, such as faults with large fault throws and uplifts of salt bodies, bring challenges for the clinoform delineation. We further introduce the structural smoothing layer at 1/4 scale on the decoder module to enhance the

structural features along the seismic structural orientations, thus filling holes and eliminating outliers in the final segmentation results.



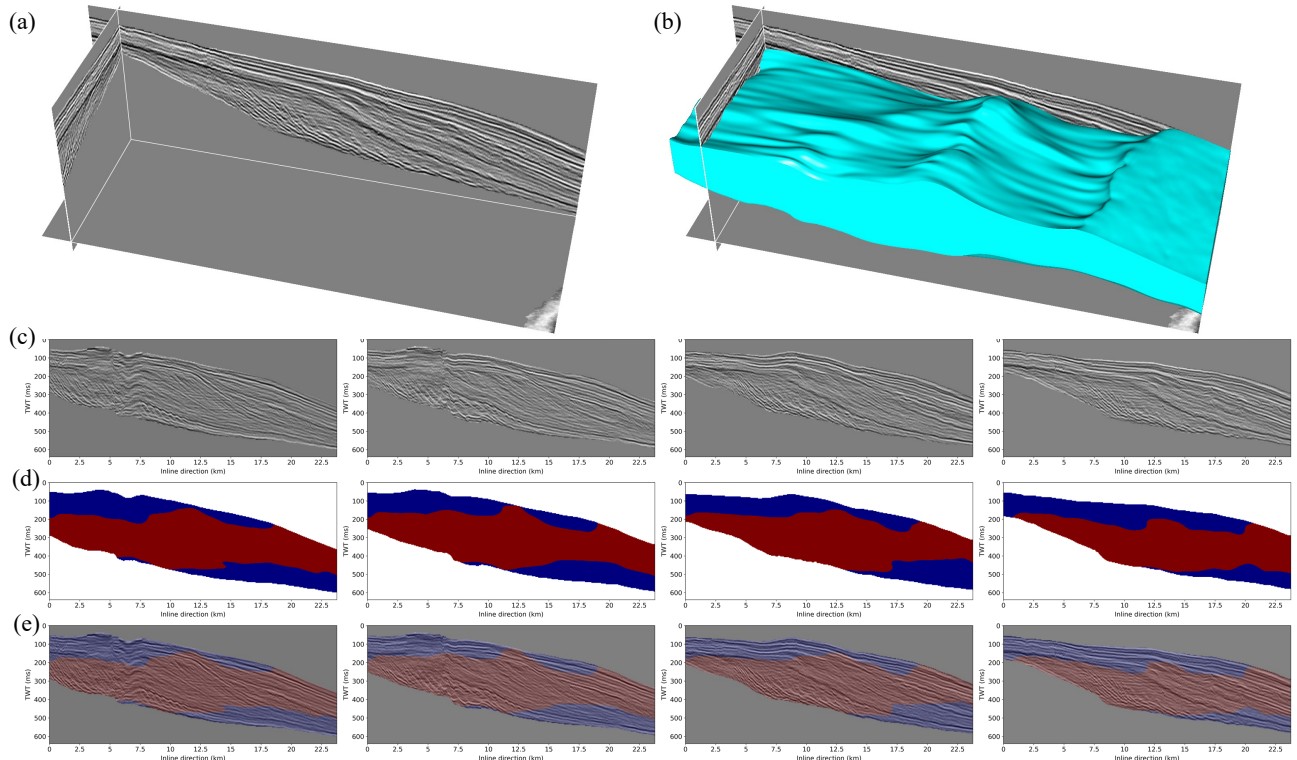

**Figure 12.** Case study one: Netherlands offshore F3 Block seismic data. (a) The 3D subset of the F3 Block seismic data, (b) 3D body of the foreset area of the predicted clinoform, (c) four 2D seismic profiles extracted from the 3D seismic volume, (d) the predicted results using the network trained with only synthetic clinoform seismic data, (e) and the results overlaid with the seismic profiles. The foreset areas of the clinoform in 2D profiles are filled with red, whereas the topset and bottomset areas of the clinoform are filled with blue. From the predicted results, the topset, foreset, and bottomset parts of the clinoform can maintain clean and complete and the boundaries between the topset and foreset and between foreset and bottomset can maintain continuity.

Fig. 12c shows some 2D seismic slices that are extracted from the 3D seismic volume and Fig. 12d displays the corresponding clinoform segmentation results. The red regions represent the foreset areas of the clinoform predicted by the trained network, whereas the blue regions represent the topset or bottomset areas of the clinoform. In order to intuitively evaluate the predicted

results, we also display the results by overlaying the predicted results with the seismic images shown in Fig. 12e. To visualize the predicted results for the entire 3D volume, we also use the classic marching cubes algorithm proposed by Lorensen and Cline (1987) to extract a 3D foreset equivalence surface as shown in Fig. 12b, where the foreset areas of the clinoform are displayed as a 3D geobody with cyan color.

As is shown in Fig. 12b,d,e, the prediction segmentation results can maintain high integrity and continuity and separate the

foreset, topset, and bottomset of the clinoform in seismic profiles, which demonstrates the reliable generalization of the trained network in this field data example. From the seismic images (Fig. 12c), they are complicated by some special structural fea-



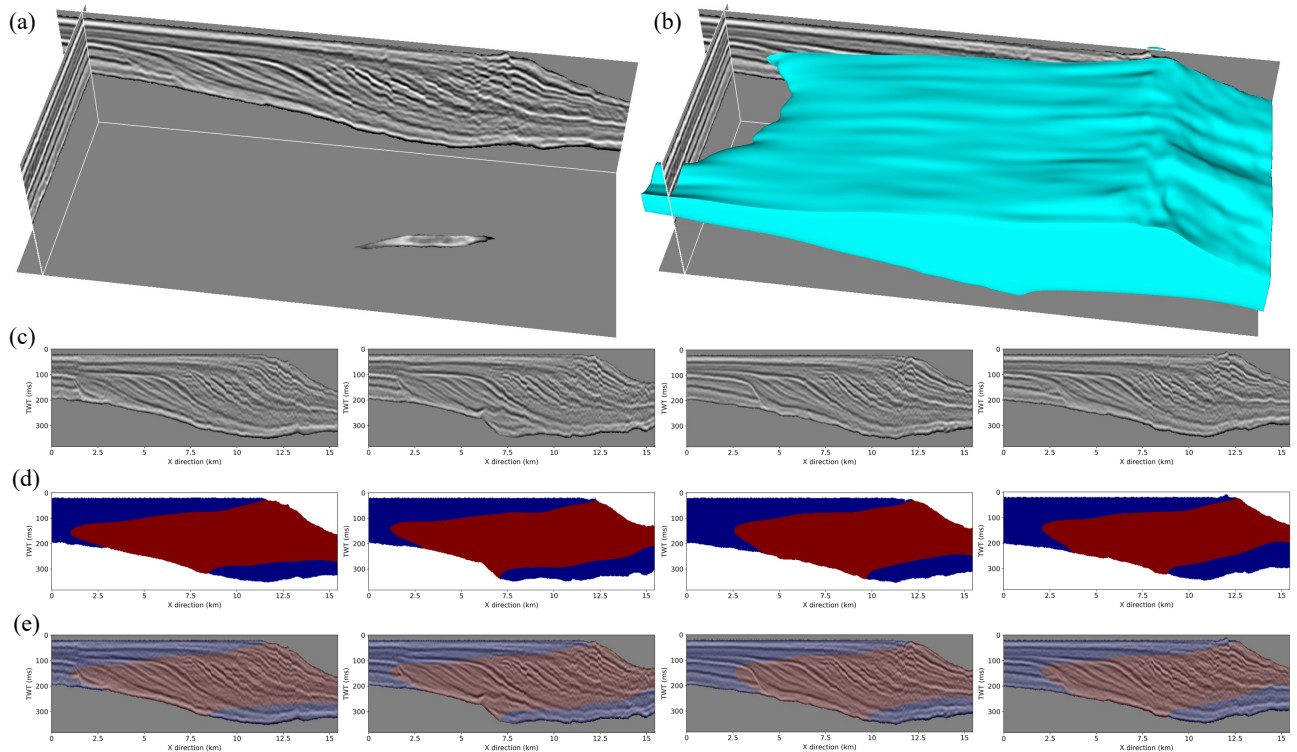

**Figure 13.** Case study two: Australia Poseidon 3D seismic data. (a) The flattened 3D subset of Australia Poseidon seismic data, (b) the 3D body of the foreset area of the predicted clinoform, (c) four randomly selected 2D seismic images, (d) the predicted results using the trained network, (e) and the results overlaid with the seismic images. The foreset areas of the clinoform in 2D profiles are filled with red, whereas the topset and bottomset areas of the clinoform in 2D profiles are filled with blue.

tures, such as faults with large fault throws, uplifts of salt bodies, and fold deformation, which are not included in the training data sets. However, our prediction results are generally not affected by these special features. This indicates that the trained network has successfully learned to analyze the geological structure information of the topset, foreset, and bottomset parts of the clinoform from a global perspective, not just from the local characteristics of seismic data. Besides, it shows that the trained network has certain anti-noise ability and generalization for some complex structures.

## 4.2 Case study two: Australia Poseidon seismic data

The second field example is the Poseidon 3D seismic volume acquired in the Browse Basin's shelf margin, North West Australia (Liu, 2018; Dixit and Mandal, 2020). The survey area of this seismic volume is 64km in the inline direction, 63km in the crossline direction, and 6000ms in the two-way travel time direction. The grid size of this 3D seismic volume are



18.75m(inline) × 12.5m(crossline) × 4ms(time). We first resample this 3D seismic volume and then select a 3D subset (618[inline] × 288[crossline] × 192[time] samples) of the Australia Poseidon 3D seismic volume as the study seismic volume.

We first extract the top and bottom surfaces of the part of clinoform deposition in the 3D seismic volume. Then we flatten
the 3D seismic volume with the top surface as a horizontal datum to eliminate the tectonic influence of the overlying strata. After flattening the 3D seismic volume, we cut the flattened seismic volume (Fig. 13a) between the two surfaces to preserve the clinoform deposits and remove the non-clinoform deposits. After these processes, we fed the 3D sub-volume slice by slice in the $y$-direction into the trained network to delineate the clinoform. To be consistent with the synthetic seismic data, each 2D slice (618[inline] × 192[time] samples) is normalized before fed into the network. Similar to study case one, we also introduce
the structural smoothing layer in the decoder module of the network to enhance the features maps along the seismic structural orientations.

We randomly select four 2D seismic slices (Fig. 13c) from the 3D seismic sub-volume and display the corresponding clinoform segmentation results in Fig. 13d. The red and blue regions represent the foreset areas of the clinoform and topset or bottomset areas of the clinoform predicted by the trained network, respectively. We also overlay the predicted results with
seismic images shown in Fig. 13e and display a 3D view of the segmented clinoform body in Fig. 13b. As shown in Fig. 13b,d,e, the predicted results are clean and complete, where the topset, foreset, and bottomset areas of the clinoform are mostly reasonably segmented.

### 4.3 Case study three: Alaska North Slope

The third field seismic data is acquired at the Alaska North Slope which is a well-known Lower Cretaceous clinoform depositional sequence. It is also a large depositional scale ($600 - 1000m$ depositional thickness in the north and $1700 - 2000m$ depositional thickness in the south), constructional, siliciclastic clinoform (Bird and Molenaar, 1992; Houseknecht et al., 2009; Ramon-Duenas et al., 2018). This study includes three publicly available, 1974-1981 vintage, 2D seismic profiles from the United States Geological Survey (USGS), which are 58-75 trackline, 28-81 trackline, and 39-81 trackline, respectively. We
select 110km[inline] × 1800ms[time], 55km[inline] × 800ms[time], 50km[inline] × 800ms[time] 2D seismic sub-profiles, respectively.

We extract the top and bottom layers of the part of clinoform deposition and another top layer of the part of a horizontal deposition in each seismic profile. Then we cut out each sub-profile between the exacted top and bottom layers and flatten it based on another top layer. We further perform gain and filtering operations on the flattened seismic images to enhance the
seismic features and the processed images are shown in the left column of Fig. 14. In segmenting clinoform in these seismic profiles, the low signal-to-noise ratio of the data and the existence of complex geological structures such as faults with large fault throws and fold deformation, bring great challenges to the network trained with only synthetic data. To reduce the impact of these complex structures on clinoform delineation, we introduce the structural smoothing layer to enhance the structural features along the seismic structural orientations.



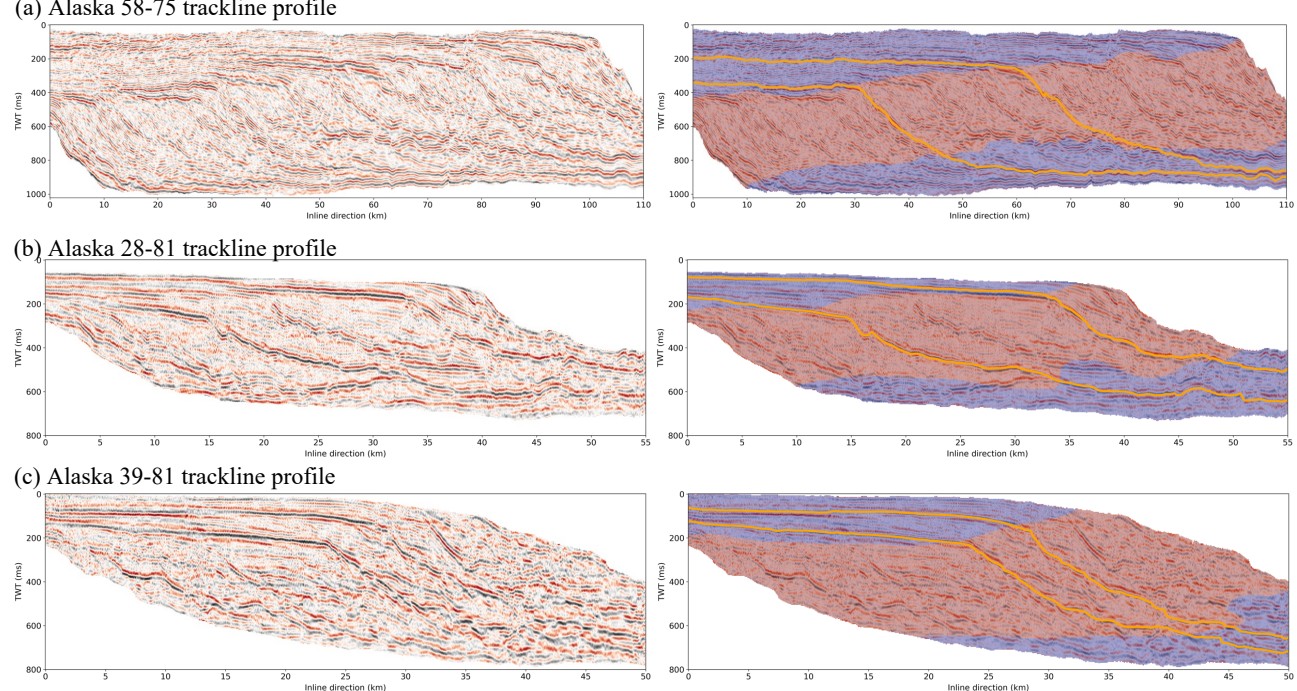

**Figure 14.** Case study three: Alaska North Slope 2D seismic data. (a) The 58-75 trackline sub-profile and predicted result. (b) The 28-81 trackline sub-profile and predicted result. (c) The 39-81 trackline sub-profile and predicted result. The survey area of these three seismic profiles are 110km, 55km, and 50km in the inline direction and 1000ms, 800ms, and 800ms in the two-way travel time direction, respectively. The seismic profiles shown in the left column are processed by flattening, gain, filtering, and normalization operations, and the segmentation results are displayed in the right column, where the foreset areas of clinoform are filled with red and the topset and bottomset areas of clinoform are filled with blue. We also display two synchronous horizons extracted in each seismic profile with orange lines.

As shown in the right column of Fig. 14, the segmentation results predicted by our modified network are clean and complete. The red regions represent the foreset areas of the clinoform and the blue regions represent the topset and bottomset areas of the clinoform. To further validate the segmentation results, we extract two synchronous horizons in each seismic profile and display them with orange lines in the right column of Fig. 14. Combining synchronous horizon distribution and segmentation results, our network can successfully segment regions with sharp slope angles on the synchronous horizons. It is indicated that our network can learn to analyze the structural features of the clinoform and can automatically segment regions with sharp slope angles (foreset regions). Besides, the introduction of the smoothing structural layer enables the network to enhance the feature features and reduce the generation of holes or outliers caused by the complex structures and low signal-to-noise ratio of the data.





## 5   Discussion

Applications on the synthetic data and three different field seismic data indicate that our training datasets generated by geological and geophysical forward modeling are geologically reasonable to train our network for accurate clinoform delineation. The introduction of the structural smoothing layer at different scales enables the trained network to enhance the structural features along the seismic structural orientations in complex field seismic data applications, thus filling holes and eliminating outliers in the final segmentation results. From the segmentation results of field seismic data, the topset, foreset, and bottomset areas of

the clinoform are accurately segmented into different classes and each class maintains high integrity and continuity.

Although the segmentation results are visually reasonable in general, some artifacts or inaccurate predictions still appear in some local areas of the results which indicates some limitations of our method. The main limitation remains in the workflow of generating synthetic training datasets by geological and geophysical forward modeling. We have not yet included any folding and faulting structures, salt bodies, or channels in generating the stratigraphic models and the corresponding seismic data.

These structure features, however, may significantly affect the seismic stratigraphic interpretation tasks. In this paper, we have flattened the seismic data to remove the folding and faulting structures before the clinoform segmentation, which makes the data structurally more consistent with the training seismic data and therefore reduces the segmentation challenges due to various structures. In the future, we need to build more realistic stratigraphic models with various structural patterns and geobodies to better train a network for interpreting more complex seismic data. In addition, the porosity model generated by SFM models

actually only considers the mechanical compaction of the sediment, which also leads to less realism of the synthetic seismic data.

In terms of network selection, we select U-net, DeeplabV3+, and DeeplabV3+ without a skip connection for the comparison test. We train these networks using the cross entropy (CE), mean squared error (MSE), binary cross entropy (BCE), and binary cross entropy with log (BCEWithLogits) loss functions, respectively. These network tests show that the DeeplabV3+ network

without skip connection using cross-entropy loss function is the most stable and accurate in both synthetic and field seismic data applications. In addition to the network architectures and loss functions, the size of the seismic data or the scale of the clinoform is another important aspect affecting the segmentation results. For example, the training dataset with a scale size of 1600×256 pixels performed well on the large-scale Alaska North Slope data but performed poorly on the small-scale F3 Block and Australia Poseidon seismic data. Therefore, we also simulated small-scale clinoform training data sets with a scale size of

900×256 pixels for F3 Block and Australia Poseidon seismic data and obtained stable and accurate segmentation results by the trained network.

In practice, the quality of the field seismic data may not be as high as synthetic data because of noise, seismic migration artifacts, and unclearly imaged reflections. We have added noise, extracted from field seismic data, to the synthetic data to train our network. However, the network may still generate holes and outliers in delineating clinoform in poor-quality seismic data.

Based on the prior geologic knowledge that each part (topset, foreset, and bottomset) of the clinoform segmentation should be a complete and continuous block, we introduce the structural smoothing layer to our network. The structure-oriented convolution kernels in the smoothing layer smooth the feature maps computed in the network along the seismic structure orientations to





suppress noisy features while enhancing the continuity of effective features in the maps, thus filling holes and eliminating outliers in the final segmentation results. Besides, the structural smoothing layer can be integrated into the network at different

scales. This means that the structural smoothing layer is highly scalable and can be introduced at any depth of the network.

Considering the computational cost, the synthetic training datasets used for network training are two-dimensional, and the initial topography used for the stratigraphic forward modeling is a simple slope surface with lateral consistency. In the future, we can upgrade our workflow from 2D to 3D by designing more complex and extensive initial topographies to generate diverse 3D clinoform seismic datasets. Additionally, we can take into account some other factors in building the porosity model to

enhance its diversity and realism. For example, we can introduce the constraints of the paleo-depth model based on the sorting effect of the ocean on sediments, and we can also introduce the constraints of the relative geologic time model according to the different compositions of sedimentary strata in different simulated times. The introduction of these multiple constraints can improve the diversity of the porosity model, enhance the realism of the synthetic data, and improve the process of the stratigraphic forward modeling. In the validation stage of the field seismic data, we can further introduce some other geophys-

ical data information (e.g., log data) to better validate the segmentation results of field seismic data. Moreover, although we only use such a workflow to solve the clinoform delineation problem at present, we can easily extend this workflow for more geological and geophysical scenarios in the future, such as sequence boundary identification, synchronous horizon extraction, relative geologic time estimation, sedimentary facies analysis, unconformity identification, etc.

## 6 Conclusions

We propose a workflow to automatically generate synthetic seismic data of clinoform and corresponding clinoform labels. In this workflow, we employ stratigraphic forward modeling to obtain numerous stratigraphic models with clinoform layers by randomly but properly choosing initial topographies, sea level curves, thermal subsidences, and rainfall patterns. Then we convert the simulated stratigraphic model into the impedance model by using a velocity-porosity relationship, while calculating

the slope model from the corresponding depth model of clinoform layers. We further perform depth-to-time conversion to the impedance model, convolve it with a Ricker wavelet with a random peak frequency and add real noise to generate a synthetic seismic image. Finally, we obtain the corresponding clinoform label according to a slope threshold. In this way, we automatically generate 1000 large-scale and 2000 small-scale synthetic seismic data and corresponding stratigraphic labels, and make these datasets available to the public.

We use an encoder-decoder network modified from DeepLabV3+ to perform clinoform delineation. In the modified network, the encoder module consists of DCNN and ASPP modules, which are mainly used to extract features at different resolutions, and the decoder module consists of a few simple convolutional and upsampling layers, which are mainly used to refine the segmentation results. Considering the complex geological structures of the field seismic data, we introduce the structural smoothing layer in the decoder module to enhance the structural features along the seismic structural orientations, thus filling

holes and eliminating outliers in the final segmentation results.

We train the modified network with only synthetic seismic datasets and then validate its performance on both synthetic and field seismic data. The segmentation results are clean, continuous, and visually reasonable. This indicates that the proposed workflow of using synthetic seismic data to train the network and using the trained network for automatic and fast clinoform delineation is plausible. Moreover, our workflow can obtain other stratigraphic labels (in addition to clinoform labels) during the geological and geophysical forward modeling. Therefore, our workflow can be easily extended for other seismic stratigraphic interpretation tasks such as sequence boundary identification, synchronous horizon extraction, shoreline trajectory identification, sedimentary facies analysis and so on.

*Code and data availability.* The synthetic seismic dataset used for training and validating our network have been uploaded to Zenodo and are free available at https://doi.org/10.5281/zenodo.7122471 (Gao et al., 2022a). The source code for the geological and geophysical forward modeling and the neural network have been uploaded to Zenodo and are free available at https://doi.org/10.5281/zenodo.7123934 (Gao et al., 2022b).

*Author contributions.* XW initiated the idea of geological and geophysical forward modeling and advised the research on it. JZ and XW initiated the idea of deep learning for seismic clinoform delineation and advised the research on it. HG conducted the research and implemented the geological and geophysical forward modeling algorithms. ZB and HG tested and modified the code for the neural network in PyTorch. HG prepared the training dataset and carried out the experiments for both synthetic and field seismic data. XW initiated the idea of introducing the structure-oriented smoothing layer in the network. XW, JZ, and XS advised on the simulation process and result analysis from a geological perspective. HG and XW prepared the paper, with contributions from all co-authors.

*Competing interests.* The authors declare that they have no conflict of interest.

*Acknowledgements.* This research is financially supported by the National Natural Science Foundation of China (grant nos. 41974121 and 42050104)





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
