# Peer review of "ClinoformNet-1.0: stratigraphic forward modeling and deep learning for seismic clinoform delineation"

_Geoscientific Model Development, 2022_

## Author Comment (AC1)

Responses to comments from reviewers

To reviewer 1:

Dear Dr. Xuesong Ding,

We sincerely appreciate all your careful reviewing so that we could get the reviewed manuscript promptly. We appreciate all your valuable comments and suggestions, which help a lot to improve our manuscript. We have corrected the figures caption and spelling mistakes and made modifications in the manuscripts according to your comments. Below we are trying to responses all your comments, suggestions, and questions. Let's discuss more if some of our explanations in the responses are not clear to you. The related modifications are not shown in the responses but are all marked in the manuscript revision history.

Thanks!

● **Specific comments:**
1. In your stratigraphic forward modeling, the thermal subsidence would change the slope of strata temporally. When you calculate the slope model, did you use the final model outputs from pyBadlands or the temporal outputs? If you are using the final model outputs, the slope of strata cannot reflect the actual slope when sediments are deposited. Do you think this would affect your automatic interpolation?

    Thank you for raising this important point. As shown in Fig. 1, we have calculated two slope models and corresponding clinoform segmentation labels based on two different model outputs (final outputs and temporal outputs). We can find that the difference between the two different slope models is not significant, probably because what we use to calculate the slope models is the local neighborhood depth value, where the local difference in thermal subsidence is not obvious. Besides, the temporal slope model is difficult to obtain for the field data in practice, thus we generally use the final model outputs to calculate the slope model. However, we also think the differences between the two slope models will have some influence on the results of seismic clinoform delineation, so we have added this consideration to the later discussion (Lines 449-451).

[Figure]

**Figure 1**. Two different slope models and corresponding clinoform segmentation labels calculated from final outputs and temporal outputs.

2. You added a structure-oriented smoothing layer to eliminate the holes or outliers. In some cases, for example with a sudden change in relative sea level or sediment supply, a sharp sequence boundary could be generated. Would the smoothing layer smooth out the sharp boundary between topset and foreset?

Thanks for your insightful comments. As shown in Figure. 10, 11 in the manuscript, the introduction of the structure-oriented smoothing layer has a minimal smoothing effect on the shape of the boundary between topset and foreset while eliminating the holes or outliers. Our structure-oriented smoothing is anisotropic and spatially varying, follow structures (Fig. 2). When a sudden change in relative sea level or sediment supply, the sharp boundary between topset and foreset could be generated (Fig. 2a, b). In this case, the structure-oriented smoothing layer will have no significant effect on the overall shape of the boundary between topset and foreset, except for the smoothing effect on the corners as shown in Fig. 2c.

[Figure]

**Figure 2**. (a) Cartoon diagram of the stratigraphic layers with a sudden change in relative sea level or sediment supply. The black and red lines represent the stratigraphic layers and boundary between topset and foreset, respectively. The orange ellipse represents the structure-oriented smoothing filter is anisotropic and the smoothing effect is more powerful in the major axis than in the minor axis. (b) The sketch of the boundary between topset and foreset. (c) The sketch of the boundary between topset and foreset after adding the structure-oriented smoothing layer.

● **Technical comments:**
1. Line 95: Change "deposition processed" to "deposition processes".
   Thanks for your suggestion. Corrected (Line 96).

2. Line 97: Change "Pybadlands" to "PyBadlands".
   Thanks. Corrected. We have changed "Pybadlands" to "PyBadlands" (Line 98).

3. Line 101: "degrades" or "retrogrades"?
   Thanks for your advice. We have modified the term to "retrogrades" (Line 106).

4. Line 102: Move "The rate of accommodation change reflects the ..." to Line 100, before "When ...".

Thanks for your suggestion. We have modified the order of the related sentences (Lines 101-106).

5. Line 104: "... is mainly related to the erosion of source domain determined by rainfall patterns, ...".
Thanks for your suggestion. We have modified the related sentence according to your advice (Line 103).

6. Figure 2 caption: Change "The workflow is used to generate ..." to "Workflow of generating ...". In the second black box, the SFM also produce depth information, so the dimension should be (X, Y, Z, N) right?
Thanks for your suggestion. We have corrected the dimension to (X, Y, Z, N).

7. Figure 3 caption: I recommend to also cite Mckenzie (1978) for the thermal subsidence curve.
Thanks for your suggestion. We have added this reference citation here.

8. Line 108: Change "100m x 100m" to "100 m x 100 m". Add a space between the number and the unit. Same applies to everywhere else.
Thanks for your advice. Corrected.

9. Line 117-118: Could the narrow width in y direction guarantee no boundary effects on your model results?
Thanks for your comment. To reduce the boundary effects on model results, we extract 2D profiles in the middle of the y-direction to build the training datasets. Besides, we have simulated the model results based on different widths (y=10 km, 20 km, 40 km, 80km) as shown in Fig. 3. According to the model results, the increase of the width of y-direction reduces the boundary effect on the model results. However, the long width of the y-direction causes a dramatic increase in computational time and memory. Therefore, we need to trade off the reasonableness of the simulation results in the sedimentary basin and the computational time and memory, and finally set the width to 40km.

[Figure]

**Figure 3**. Four 2D examples of the model results based on different widths (y =10 km, 20 km, 40 km, 80 km) from stratigraphic forward modeling.

10. Line 130: Remove "in simulation".

Thanks. Corrected. We have removed the phrase "in simulation" in this sentence (Line 131).

11. Line 135-136: Change "... because precipitation can ... to the sedimentary basin" to "... by changing the power of river streams on eroding and transport sediments to sedimentary basins".

Thanks for your suggestion. We have modified the related sentence according to your advice (Lines 136-137).

12. Line 138: What matrix do you use to measure a stable sediment supply? Why a stable sediment supply is required?

Thanks for your comments. We use the sediment flux and the size of the final simulation model as references for measuring a stable sediment supply. As shown in Fig. 4, we display two different simulation results with and without uplifting the mountain range during the simulation processes. Compared to the original simulation model (Fig. 4a), the simulation model without uplifting (Fig. 4b) has less sediment flux, and the peak of sediment flux decreases significantly in the later stages compared to the earlier stages. In addition, the uplift of the mountain can increase the sediment flux and reduce the variation of the sediment flux during long-period erosion.

When the sediment supply is unstable during the simulation processes, it may lead to deposition hiatus shown in Fig. 3a or large changes in the size of the stratigraphic models due to large changes in sediment supply during the simulation. These conditions can lead to low consistency of the synthetic datasets, which is not conducive to the network training and prediction.

[Figure]

**Figure 4**. Simulation results of adding and not adding the uplift of the mountain

13. Line 147: I believe pyBadlands also produces paleo-depth in the outputs.

Thanks for your comment. We have modified this ambiguous sentence to "We further use the attributes of relative geologic time, paleo-depth, and porosity on the depth curves to interpolate the corresponding full models as shown in color in each column of Fig. 4" (Lines 147-150).

14. Line 262: Remove the second "by".

Thanks. Corrected. We have removed the second "by" in this sentence (Line 262).

15. Line 368-369: "The red and blue regions ..." is a repeat. You can also remove this sentence at Line 391-392.

Thanks for your suggestion. Corrected.

16. Line 372: Did you compare with previous interpretation or just eyeballed?

Thanks for your constructive comment. We have found some interpretation results by some authors. Taking the Alaska North Slope 58-75 profile as an example, our prediction result is closer to the human interpretation result, especially for boundary between topset and foreset. We have added this comparison to further demonstrate the performance of our method in our manuscript (Lines 392-393).

17. Line 396: Remove the "feature" at the end.

Thanks. Corrected.

18. Line 402: How to determine the structural smoothing layer?

Thanks for your comment. We have introduced how to determine the structural smoothing layer in lines 299-303 in the manuscript. "We first estimate the structural orientation at each pixel of the input seismic image by using the method of structure tensor (Van Vliet and Verbeek, 1995; Fehmers and Höcker, 2003; Hale, 2009). Then we construct the corresponding smooth convolution kernels along the seismic structures based on the estimated orientations. Finally, we apply these structure-oriented smoothing kernels to the feature maps at the scales of 1/4 and 1/16 in the network." (Lines 300-304)

19. Line 409: Change "generating the stratigraphic models" to "generating the stratigraphic layers".

Thanks for your suggestion. Corrected. We have replaced "models" with "layers" in this sentence (Lines 416-417).

---

## Author Comment (AC2)

To reviewer 2:

Dear Dr. Mark Jessell,

We would like to thank all your wonderful work so that we could get the reviewed manuscript promptly. We appreciate all your valuable comments and suggestions, which are helpful to improve our manuscript. In this document, we try to address your comments in detail. Let's discuss more if some of our explanations in the responses are not clear to you. The related modifications are not shown in the responses but are all marked in the manuscript revision history.

Thanks!

1.  The challenges facing the use of ML in the geosciences were nicely summarised in the following paper, that may be worth referencing. A. Karpatne, I. Ebert-Uphoff, S. Ravela, H. A. Babaie and V. Kumar, "Machine Learning for the Geosciences: Challenges and Opportunities," in IEEE Transactions on Knowledge and Data Engineering, vol. 31, no. 8, pp. 1544-1554, 1 Aug. 2019, doi: 10.1109/TKDE.2018.2861006.
    Thanks for your suggestion. We have added this reference citation in the manuscript (Line 56).

2.  May be good to refer to other works that are in the space of using synthetic labelled models to train ML:
    https://gmd.copernicus.org/preprints/gmd-2022-245/#discussion
    https://essd.copernicus.org/articles/14/381/2022/
    https://doi.org/10.1016/j.cageo.2021.104701
    https://geoscienceletters.springeropen.com/articles/10.1186/s40562-022-00241-y
    Thanks for your suggestion. We have added these related reference citations to the space of using synthetic labelled datasets to train the network in the manuscript (Lines 57-58).

3.  Although artificial geophysical noise is tested in this study, unless I am mistaken there is another aspect of model variability, namely the natural variability of rock properties, which is not taken into account. This can be due to initial variations in provenance, or in variable compaction. This could be discussed in later sections and considered for future studies.
    Thank you for raising this important point. The diversity of rock properties due to variable compaction is reflected in the porosity model obtained from stratigraphic forward modeling. However, the diversity of rock properties due to initial variation in provenance has not been included in the simulation processes. Therefore, we have added the natural variability of rock properties to enhance the diversity and realism of the porosity model in the discussion (Lines 448-449).

4.  The use of a specific modelling engine (pybadlands) rather than an alternative could

have implications for the outcomes, in the discussion it may be worth highlighting the strengths and weakness of this particular simulation platform versus others that exist in terms of their predictions?

Thanks for your constructive comment. We have highlighted the strengths and weakness of the specific modelling engine (PyBadlands) in the discussion (Lines 410-415).